# Epidemiological Surveillance of Eye Disease and People Awareness in the Abruzzo Region, Italy

**DOI:** 10.3390/medicina57090978

**Published:** 2021-09-17

**Authors:** Leonardo Mastropasqua, Rossella D’Aloisio, Alessandra Mastrocola, Fabiana Perna, Luca Cerino, Loredana Cerbara, Filippo Cruciani, Lisa Toto

**Affiliations:** 1Ophthalmology Clinic, Department of Medicine and Science of Ageing, University G. D’Annunzio Chieti-Pescara, 66100 Chieti, Italy; mastropa@unich.it (L.M.); studiomastrocola@gmail.com (A.M.); f.perna@iapb.it (F.P.); lucacerino92@gmail.com (L.C.); l.toto@unich.it (L.T.); 2Italian National Centre of Services and Research for the Prevention of Blindness and Rehabilitation of the Visually Impaired-WHOCC, IAPB Italy Onlus-FPG IRCCS, 00185 Roma, Italy; filippo.cruciani@uniroma1.it; 3Institute for Research on Population and Social Policies, Italian National Research Council (CNR-IRPPS), 00185 Rome, Italy; loredana.cerbara@irpps.cnr.it

**Keywords:** population screening, eye disease, prevalence, awareness

## Abstract

*Background and Objectives*: Vision impairments and related blindness are major public health problems. The prevalence of eye disease and barriers to optimal care markedly vary among different geographic areas. In the Abruzzo region (central Italy), an epidemiological surveillance on the state of ocular health in the population aged over 50 years was performed in 2019. *Materials and Methods*: Participants were sampled to be representative of the region’s inhabitants. Data were collected through a telephone interview and an eye examination. Prevalence of cataract, glaucoma, retinopathy, and maculopathy was assessed. The Cohen’s kappa (k) was used to measure the agreement between the presence of eye disease and awareness of the disease by the participants. *Results*: Overall, 983 people with a mean age of 66.0 ± 9.5 years were included in the study. The prevalence of cataracts, glaucoma, maculopathy, and retinopathy was 52.6%, 5.3%, 5.6%, and 29.1%, respectively. Among the total of the affected people, those aware of their condition were 21.8% (k = 0.12, slight agreement) for cataract, 65.4% (k = 0.78, substantial agreement) for glaucoma, 7.1% (k = 0.10, slight agreement) for maculopathy, and 0% for retinopathy (k = −0.004, agreement lower than that expected by chance). Refractive defects were corrected in the vast majority of participants. *Conclusion*: In the Abruzzo region, about two thirds of citizens aged 50 years or over suffer from cataract, glaucoma, retinopathy, or maculopathy, which are recognized as leading causes of blindness. Many people with eye disease do not know they have it. These data can be used by clinicians and policymakers to undertake clinical, political, and social actions.

## 1. Introduction

Vision impairments and related blindness are major public health concerns in middle-aged and elderly adults worldwide, associated with diminished quality of life and increased risk of falls and death [1,2,3].

Available estimates suggest that 36 million people were blind, and 217 million people had moderate or severe vision impairment worldwide in 2015 [4].

The number of people affected by the common causes of vision loss has increased substantially as the population increases and ages. Simultaneously, the number of people with avoidable visual loss has increased. Cataract continues to cause most cases of blindness and moderate or severe vision impairment in adults aged 50 years and older, but also glaucoma, age-related macular degeneration, and retinopathy are major problems [1,5]. 

Estimating the prevalence of vision impairment and patient needs are a fundamental basis of public health policies [5]. 

Based on the data, local healthcare systems can implement appropriate actions to address this largely preventable global problem and provide adequate eye care services.

Given these premises, a population-based epidemiological surveillance was conducted among individuals aged ≥ 50 years in the Abruzzo region (Italy), aiming at estimating the prevalence of visual impairments (i.e., retinopathy, maculopathy, cataract, glaucoma) and people’s characteristics and attitudes about eye health.

## 2. Materials and Methods

This was an epidemiological surveillance on the state of ocular health in the population aged over 50 years of the Abruzzo Region in 2019.

Participants were sampled to be representative of the region’s inhabitants aged 50 years or older (2-stage sampling of randomly selected towns and cities of different sizes in the four provinces and representative distribution by gender and age classes). The sampling frame for selection was the list of persons living in the community, based on the local population registers held by the municipalities. After stratifying for gender and age, a sampling technique with probabilities proportionate to the size of each population stratum was used.

Data were collected through telephone interviews investigating socio-demographic and anamnestic information. The following data were collected: age, gender, school education, living status, working status, lifestyle information (physical activity, smoking habit, alcohol consumption, and daily exposition to ultraviolet rays), body mass index, and chronic diseases. Information about attention paid to the general and eye health and perceived health status was also collected.

Therefore, all participants underwent an eye examination through a mobile clinic. Refractive errors using a subjective and autorefractometer evaluation (auto refractometer AR-600 Nidek, Aichi, 443-0038, Japan), corneal assessment using slitlamp biomicroscopy, Amsler grid test, Ishihara test, cover-uncover test, and tonometric assessment were performed. In addition, in all patients a color fundus retinography was acquired using KOWA Nonmyd WX-3D retinal camera (Torrance, CA, USA).

Attention was paid to the following main eye diseases: cataract, glaucoma-related optic disc modifications, advanced age-related macular degeneration (AMD) (maculopathy), diabetic or hypertensive retinopathy, or other retinal vascular diseases (retinopathy). People diagnosed with these diseases were asked if they were aware of their condition.

The study was conducted in accordance with the Helsinki Declaration on Medical Research on Humans and with the Good Clinical Practice (GCP). The study did not require approval by the Ethics Committees, since it was based on the voluntary participation of people to an epidemiological screening campaign. Data collected were anonymous. Participants signed an informed consent for data protection according to European and national legislation.

### Statistical Analysis

Descriptive data were summarized as the mean and standard deviation, median, and interquartile range or frequencies and proportions.

Characteristics of the study population were assessed overall and by awareness about eye diseases.

Groups were compared using the Student’s *t*-test (continuous, normally distributed variables), Mann–Whitney U-test (continuous, not normally distributed variables), chi-square test or Fisher exact test (categorical variables), as appropriate.

The Cohen’s kappa (k) was used to measure the agreement between the presence of eye disease and awareness of participants about the disease. A value of 1 implies perfect agreement and values less than 1 imply less than perfect agreement. Five levels of agreement have been identified based on k value: slight (0.01–0.20), fair (0.21–0.40), moderate (0.41–0.60), substantial (0.61–0.80), and almost perfect (0.81–1.00); negative k means that the agreement is less than that expected just by chance [6]. 

A *p*-value < 0.05 was considered statistically significant. Statistical analyses were performed using the SAS program, version 9.4 (SAS Institute Inc., Cary, NC, USA).

## 3. Results

Overall, 983 people were included in the study. Participants’ characteristics are summarized in Table 1.

Mean age was 66.0 ± 9.5 years, men represented 45.9% of the sample, school education level was inferior to high school for 65.6% of participants, 20.7% performed physical activity, 15.5% were smokers, 41.8% consumed alcohol regularly, 63.1% were overweight or obese, and 41.8% had hypertension. Although unhealthy lifestyle and comorbidities were documented in many patients, 78.0% declared regularly checking their own health status and 60.9% declared to attend an eye examination at least every 2 years. Good/excellent health was reported by 84.8% of participants.

In eye examination, astigmatism was identified as the most frequent refractive defect, while daltonism and strabismus were seldom detected (Table 2).

In terms of primary study objective, the prevalence of cataract, glaucoma, maculopathy, and retinopathy was 52.6% 5.3%, 5.6%, and 29.1%, respectively (Table 2 and Figure 1).

Among the total of affected people, those aware of their condition were 21.8% (k = 0.12, slight agreement) for cataract, 65.4% (k = 0.78, substantial agreement) for glaucoma, 7.1% (k = 0.10, slight agreement) for maculopathy, and 0% for retinopathy (k = −0.004, agreement lower than that expected by chance) (Figure 2).

Comparisons between characteristics of people aware or not aware was feasible for cataract only (for the other diseases, prevalence was low, and sub-samples stratified by agreement insufficient). Compared to people not aware about their cataract, aware people were older (73 vs. 67 years), were more often women than men (60% vs. 40%), were less likely to be married (23.8% vs. 9.3%) and were more likely to have a low school education (58.1 vs. 33.9%) (Table 3).

## 4. Discussion

### 4.1. Main Findings

Low vision affects many people aged over 50 years. However, in our study most individuals could see well with best corrected or uncorrected refractive errors, and only 3.3% had severe visual impairment. Cataract was frequent and affected half of the population, while retinopathy was found in almost one third. Maculopathy and glaucoma affected about 1 out of 200 people. While people reported a good compliance to regular checks of their general and eye health, this study suggested an important gap between the presence/absence of eye disease and people’s awareness about the disease. The majority of people with glaucoma (65.5%) knew they had the disease. The level of awareness substantially decreased for the other eye diseases: only 21.8% of people with cataract and 7.1% with maculopathy knew they had the disease, while nobody of 138 affected people was aware about their retinopathy.

### 4.2. Comparisons with Existing Knowledge

Causes of visual impairments were measured in many countries and settings, documenting that prevalence is influenced by ethnicity and healthcare resource availability [7,8,9,10]. Recently, an updated review from the Global Vision Database identified 288 studies of 3,983,541 participants from 98 countries. It documented that cataract and uncorrected refractive error combined contributed to 55% of blindness and 77% of vision impairment in adults aged 50 years and older in 2015 [5]. World regions varied markedly in the causes of blindness and vision impairment in this age group, with a low prevalence of cataract (<22% for blindness and 14.1–15.9% for vision impairment) and a high prevalence of age-related macular degeneration (>14% of blindness) as causes in the high-income subregions [5].

In another Italian population-based study, the main cause of eye impairment in people aged 40 years and over was unoperated cataract (34.8%), followed by glaucoma (21.7%), degenerative myopia (13.0%), and maculopathy (8.7%) [11]. Furthermore, compared to our study population, in a study conducted in the north-east of Italy and involving 1162 people aged 60 years or over, prevalence of maculopathy was markedly higher (62.7%), due to the inclusion of all stages of AMD and the different ages of the patients [12].

Poor health awareness of these conditions and their complications causes a delay in seeking medical care and precludes the chance of early intervention and prevention [5]. Therefore, raising public awareness of ocular diseases plays a significant role in the early diagnosis and treatment of such conditions, thus reducing the burden of visual impairment. To date, variable results have been reported about the level of awareness of common ocular diseases worldwide [13,14,15,16,17,18]. For example, in developed countries such as Canada, the level of awareness of ocular diseases was reported at 69% for cataract and 41% for glaucoma [19]. In India, knowledge about ocular diseases was poor both in urban and rural areas [20,21]. For instance, in a report from southern India, the majority of patients (90%) with glaucoma were not aware of the condition and its complications [22]. In another study in India, a very poor awareness of glaucoma (3.2%) and diabetic retinopathy (27%) was also reported [14].

### 4.3. Implications for Research and Clinical Practice

Vision impairment is a key issue for people’s quality of life and the public health of different countries. The burden of uncorrected refractive error contributes to almost half of the moderate and severe vision impairment burden, especially in low-income countries [22]. In high-income countries, effective diabetic retinopathy screening programs are in place. However, in Italy a standardized national diabetic eye screening program does not exist yet. On the other hand, due to their asymptomatic and/or monocular nature at early stages, in many settings glaucoma and maculopathy are not arrested or mitigated by timely interventions, although they are leading causes of blindness. 

Efforts to identify organizational and cultural barriers should be made to improve access, equity, and efficacy of eye care to prevent blindness. Local periodical surveillance or annual patient recall could be effective strategies to identify priority and address actions [5,22].

Moreover, telemedicine in ocular diagnosis needs to be implemented in primary settings worldwide.

### 4.4. Strengths and Limitations

The study has strengths and limitations. The major strength is that, to the best of our knowledge, this is the first evaluation of the awareness of eye diseases in an Italian population. Furthermore, the study population is quite representative of many central and southern Italian communities.

On the other hand, the major limitation was the small number of subgroups examined to assess the factors associated with the awareness of having glaucoma, maculopathy, or retinopathy. Nevertheless, factors associated with the knowledge of having cataract suggest the role of demographic and socio-economic characteristics as correlates of awareness.

Furthermore, given the mobile clinic setting, we could not diagnose all eye diseases (e.g., hereditary retinal diseases) or classify the stages of the diseases (e.g., strabism, glaucoma, AMD, other retinal diseases) but just detect their presence or absence. This was due to the lack of specific necessary exams or instruments, such as optical coherence tomography, electroretinogram tests, and genetic tests.

## 5. Conclusions

This regional epidemiological study provided useful information to clinicians and policymakers, highlighting that in this area, the refractive defect is corrected in the large majority of people aged 50 years or over. However, about two thirds of citizens in this age class suffer from cataract, glaucoma, retinopathy, or maculopathy, which are recognized as leading causes of blindness. Many people with eye disease do not know they have it. These data can be used to improve access to care and promote information campaigns.

## Figures and Tables

**Figure 1 medicina-57-00978-f001:**
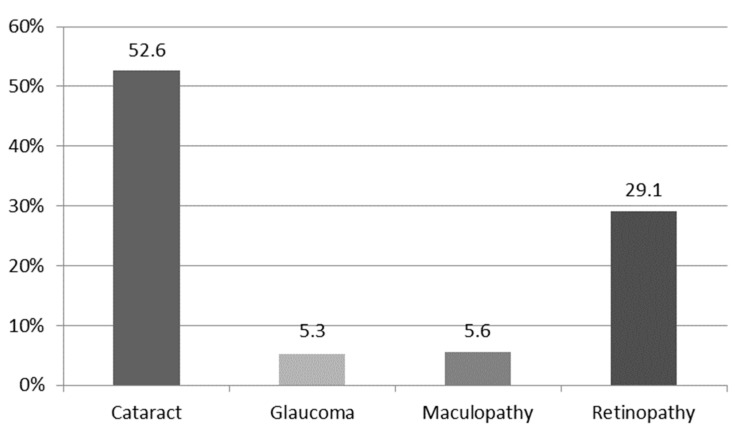
Prevalence of eye diseases among the general population aged ≥ 50 years.

**Figure 2 medicina-57-00978-f002:**
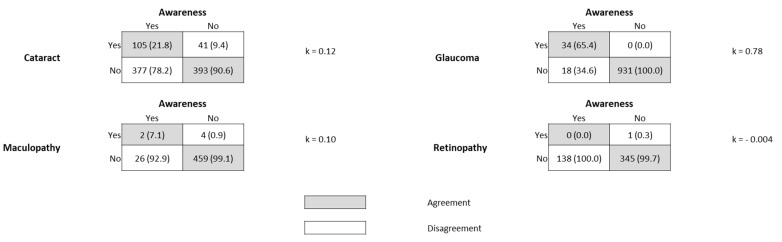
People aware about their eye disease. Rows indicate the presence/absence of eye disease whilst the columns indicate the awareness of participants about the disease.

**Table 1 medicina-57-00978-t001:** Characteristics of the study participants.

	*n* with Available Data	Mean and Standard Deviation or Frequency and Proportions
*n*		983
Socio-demographic characteristics		
Men (%)	983	451 (45.9)
Mean age (years)	983	66.0 ± 9.5
Age in classes (%):	983	
<65 years		438 (44.6)
≥65 years		545 (55.4)
School education (%):	983	
<5 years		62 (6.3)
Primary school		271 (27.6)
Secondary school		312 (31.7)
High school		280 (28.5)
University degree		58 (5.9)
Civil status (%)	983	
Married/Partner		851 (86.6)
Single/Divorced/Widow		132 (13.4)
Working status (%)	983	
Employed		230 (23.4)
Unemployed/retired		753 (76.6)
Lifestyle		
Physical activity (%)	983	
Regular		95 (9.7)
Occasional		108 (11.0)
Never		780 (79.3)
Alcohol consumption (%)	983	
Regular		411 (41.8)
Occasional		107 (10.9)
Never		465 (47.3)
Smoking (%)	983	
Yes		152 (15.5)
Ex		273 (27.8)
No		558 (56.8)
If yes, no. of cigarettes/day (%)	145	
≤10		88 (60.7)
11–20		48 (33.1)
>20		9 (6.2)
If ex, date of cessation (%)	273	
<10 years		28 (10.3)
>10 years		56 (20.5)
>20 years		64 (23.4)
>30 years		77 (28.2)
>40 years		40 (14.7)
Unknown		8 (2.9)
Daily exposition to ultraviolet rays (%)	982	
None		4 (0.4)
Low (1–2 h/day)		230 (23.4)
Intermediate (2–3 h/day)		320 (32.6)
High (>3 h/day)		428 (43.6)
Clinical characteristics		
Mean BMI (kg/m^2^)	972	26.6 ± 4.1
BMI in classes (%):	972	
Underweight		11 (1.1)
Normal weight		346 (35.7)
Overweight		430 (44.4)
Obesity		181 (18.7)
Type 2 diabetes (%)	983	130 (13.2)
Hypertension (%)	983	411 (41.8)
Cardiovascular diseases (%)	983	119 (12.1)
Neurologic diseases (%)	983	26 (2.6)
Renal diseases (%)	983	50 (5.1)
Rheumatological diseases (%)	983	234 (23.8)
Attention to the own health status		
Do you check regularly your health status? (%)	983	
Yes		767 (78.0)
Sometimes/only in case of problems		198 (20.1)
Never		18 (1.8)
How would you define your health? (%)	982	
Excellent		109 (11.1)
Good		724 (73.7)
Intermediate		142 (14.5)
Poor		7 (0.7)
Frequency of eye examination (%)	983	
More than once a year		75 (7.6)
Once a year		305 (31.0)
Every 2 years		219 (22.3)
Occasionally		384 (39.1)

**Table 2 medicina-57-00978-t002:** General eye health status and prevalence of eye diseases.

Eye Examination	*n* with Available Data	Mean and Standard Deviation or Frequency and Proportions
*n*		983
Visual perception (%)		
See well with or without glasses (20/20 snellen visual acuity)	983	550 (56.0)
Low vision (best corrected or uncorrected)	983	32 (3.3)
Refractive error (%)	983	
Myopia (>−0.25 sph)		96 (9.8)
Hyperopia (>+0.25 sph)		313 (31.8)
Astigmatism (±0.25 cyl)		435 (44.3)
None		139 (14.1)
AMSLER grid: presence of (%)	965	
Metamorphopsia		63 (6.5)
Scotoma		2 (0.2)
Scotoma and metamorphopsia		2 (0.2)
None		898 (93.1)
Discromatopsia (Ishihara test) (%)	971	38 (3.9)
Strabismus (Cover–uncover test) (%)	954	46 (4.8)
Corneal disorder (%)	919	5 (0.5)
Mean tone (mmHg)	977	14.7 ± 3.2
Cataract (%)	916	482 (52.6)
History of surgical treatment for cataract (%)	186	97 (52.2)
Satisfaction with surgical treatment for cataract (%)	97	
Very satisfied		9 (9.3)
Satisfied		83 (85.6)
Partly satisfied		5 (5.2)
Glaucoma-related optic nerve head alterations (%)	981	52 (5.3)
Advanced Age-Related Macular Degeneration (AMD)/maculopathy (%)	480	27 (5.6)
Vascular eye diseases (diabetic retinopathy, hypertensive retinopathy, occlusive vasculopathies)/retinopathy (%)	474	138 (29.1)

**Table 3 medicina-57-00978-t003:** Subgroup of patients with cataract (*n* = 482) and their characteristics by awareness about their cataract.

	Aware	Not Aware	*p*-Value *
*n*	105	377	
Socio-demographic characteristics			
Men (%)	42 (40.0)	172 (45.6)	0.30
Mean age (years)	73.0 ± 6.8	67.0 ± 8.4	**<0.0001**
School education (%)			
<Secondary school	61 (58.1)	128 (33.9)	**0.0005**
≥Secondary school	44 (41.9)	249 (66.1)	
Civil status:			
Married/Partner	78 (74.3)	333 (88.3)	**0.0003**
Single/Divorced/Widow	27 (25.7)	44 (11.7)	
Working status (%)			
Retired	88 (83.8)	245 (65.0)	**<0.0001**
Employed	17 (16.2)	132 (35.0)	
Lifestyle			
Physical activity (%)			
Regular	3 (2.9)	28 (7.4)	0.16
Sometimes	9 (8.6)	42 (11.1)	
Never	93 (88.6)	307 (81.4)	
Alcohol consumption (%)			
Regular	37 (35.2)	160 (42.4)	0.15
Sometimes	9 (8.6)	45 (11.9)	
Never	59 (56.2)	172 (45.6)	
Smoking (%)			
Yes	10 (9.5)	62 (16.4)	0.18
Ex	29 (27.6)	106 (28.1)	
No	66 (62.9)	209 (55.4)	
Clinical characteristics			
Mean BMI (kg/m^2^)	27.1 ± 4.7	26.7 ± 4.0	0.95
BMI in classes (%):			
Underweight	0 (0.0)	4 (1.1)	0.78
Normal weight	37 (36.3)	123 (33.2)	
Overweight	43 (42.2)	170 (45.8)	
Obesity	22 (21.6)	74 (19.9)	
Type 2 diabetes (%)	16 (15.2)	60 (15.9)	0.87
Hypertension (%)	62 (59.0)	158 (41.9)	**0.002**
Cardiovascular diseases (%)	19 (18.1)	45 (11.9)	0.10
Neurologic diseases (%)	2 (1.9)	11 (2.9)	0.74
Renal diseases (%)	8 (7.6)	25 (6.6)	0.72
Rheumatological diseases (%)	27 (25.7)	106 (28.1)	0.63
Attention to the own health status			
Do you check regularly your health status? (%)			
Yes	85 (81.0)	299 (79.3)	0.37
Sometimes/only in case of problem	20 (19.0)	71 (18.8)	
Never	0 (0.0)	7 (1.9)	
How would you define your health? (%)			
Excellent	2 (1.9)	34 (9.0)	**0.003**
Good	76 (72.4)	286 (76.1)	
Intermediate	26 (24.8)	55 (14.6)	
Poor	1 (1.0)	1 (0.3)	
Frequency of eye examination (%)			
More than once a year	9 (8.6)	26 (6.9)	0.21
Once a year	39 (37.1)	104 (27.6)	
Every 2 years	22 (21.0)	91 (24.1)	
Occasionally	35 (33.3)	156 (41.4)	
**Other eye disease**			
Glaucoma (%)	8 (7.6)	8 (2.1)	**0.01**
Maculopathy (%)	3 (2.9)	5 (1.3)	0.38
Retinopathy (%)	1 (1.0)	0 (0.0)	0.22

* Student’s *t*-test, Mann–Whitney U-test, chi-square test or Fisher exact test, as appropriate. Statistically significant *p*-value (<0.05) are in bold text.

## Data Availability

The study database is not publicly available. It could be provided upon reasonable request to the authors.

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
