# Peer review of "Epidemiological Surveillance of Eye Disease and People Awareness in the Abruzzo Region, Italy"

_medicina, 2021, doi:10.3390/medicina57090978_

Round 1
Reviewer 1 Report
I have any comments and suggestions
Author Response
#Reviewer 1
I have any comments and suggestions

Reviewer 2 Report
Explain how do you determined the degree of daily exposure to UV light.
I recommend to divide vascular retinal diseases in subgroups (example; DR, occlusive vasculopathies...)
Explain the lack of data about optic nerve diseases, hereditary retinal diseases, amblyopia...
Author Response
#Reviewer 2
Explain how do you determined the degree of daily exposure to UV light.
The specification was reported in Table 1. We have better specified that the questionnaire assessed the average number of hours in a day spent outdoor.
I recommend to divide vascular retinal diseases in subgroups (example; DR, occlusive vasculopathies...).
In the database, the different vascular retinal diseases were reported as one single class without the possibility to distinguish the different subgroups.
Explain the lack of data about optic nerve diseases, hereditary retinal diseases, amblyopia...
Given the mobile clinic setting, we couldn’t diagnose all diseases, for example hereditary retinal diseases, because of the lack of specific necessary exams such as OCT and electroretinogram tests. Regarding optic nerve diseases we only considered glaucoma-related optic nerve head alterations following JB Jonas’s optic disc morphometry classification. We have added this issue among the study limitations.

Reviewer 3 Report
The authors should check the consistency of their data and improve the presentation of their data
- The sum of smokers 152 is not consistent with the following data of the daily cigarette consumptions which in total are 145 following their data
- The total daily ultraviolet exposure results in 982 persons
- The weight from underweitght to obesity results in 968 persons were are the missing ?
- The “how do you define your health results in 982 persons not 983
- Figure 1 legends to small should be larger, Numbers very small should be larger
- How was the presence of cataract classified in slit lamp examination? LOCS III? Or any visibility of the lens and its structures??
- Table 2 criteria for the differentiation of :
- Reading well
- Visually impaired or reduced vision
- Myopia (which extend from – 0,25 sph? Or larger than -1dpt??) hyperopia same question larger than + 0,25 sph?? Or >+1,5 dptr? Etc
- Metamorphopsia how qualified?
- Strabism which extend of angle lead to the diagnosis?
- What was the classifier for Glaucoma related optic nerve head alterations
- The same for AMD and vascular eye disease?
Fig. 2 How were the 4 field tables filled? This should be explained row 1= answer to personal question and row 2: result from examination
Table 3 is not clear. What are the significances indicating? The presence of awareness compared to the result of examination? If yes than the data of personal awareness and non awareness should be presented together with the data on item present or not present in examination. The appropriate test would be a Fisher contingency .
The discussion is appropriate but to be certain of the right conclusions the data presentation and evaluation and especially the limits of decisions to approve or disapprove a pathological finding must be elucidated.
Author Response
#Reviewer 3
The authors should check the consistency of their data and improve the presentation of their data
In general, some data cannot be referred to the entire sample because of missing data. Hence the slight differences found. We have added a column in tables 1 and 2 to report for each variable the actual number of valid values.
- The sum of smokers 152 is not consistent with the following data of the daily cigarette consumptions which in total are 145 following their data
The number of smokers is equivalent to the total people in the sample that actually smoke, but some of these (152-145=7) do not smoke cigarettes.
- The total daily ultraviolet exposure results in 982 persons
Missing data: one person did not provide the answer.
- The weight from underweitght to obesity results in 968 persons were are the missing?
Unfortunately, some respondents did not want to provide weight and height for the determination of the level of obesity. Actually, these data were not mandatory.
- The “how do you define your health results in 982 persons not 983
Missing data: one person did not provide the answer.
- Figure 1 legends to small should be larger, Numbers very small should be larger
Done. Legends and numbers have been adapted to the figure
- How was the presence of cataract classified in slit lamp examination? LOCS III? Or any visibility of the lens and its structures??
The presence of cataract was classified based on LOCS III system.
- Table 2 criteria for the differentiation of :
- Reading well
- Visually impaired or reduced vision
- Myopia (which extend from – 0,25 sph? Or larger than -1 dpt??) hyperopia same question larger than + 0,25 sph?? Or >+1,5 dptr? Etc
- Metamorphopsia how qualified?
- Strabism which extend of angle lead to the diagnosis?
- What was the classifier for Glaucoma related optic nerve head alterations.
- The same for AMD and vascular eye disease?
Criteria for diagnoses have been reported in table 2 if available. Table 2 has been also simplified due to the lack of possibility to classify in a standard way all the items (due to the clinical mobile setting, just presence/absence of the diseases could be recorded), e.g.:
- Strabism was detected through the cover/uncover test to verify the presence of the condition, but the angle was not evaluated.
- As for glaucoma, we were able to detect only glaucoma-related optic nerve head alterations following JB Jonas’s optic disc morphometry classification.
- As for AMD, we did not evaluate the stage of ocular diseases but only considered if present or not. However, the following were the considered classification: New Beckman staging for age-related macular degeneration. UK classification for DR.
We have added this issue among the study limitations.
Fig. 2 How were the 4 field tables filled? This should be explained row 1= answer to personal question and row 2: result from examination
In each table of Figure 2 the rows indicate the presence/absence of eye disease whilst the columns indicate the awareness of participants about the disease. In the amended figure 2, this was better explained in the title of the figure.
Table 3 is not clear. What are the significances indicating? The presence of awareness compared to the result of examination? If yes than the data of personal awareness and non awareness should be presented together with the data on item present or not present in examination. The appropriate test would be a Fisher contingency.
Table 3 contains only the 482 people with cataract divided in two subgroups: those aware and those not aware of their cataract. The characteristics of the two groups were formally compared with statistical tests already explained in the statistical section of the manuscript (Groups were compared using Student’s t-test (continuous, normally distributed variables), Mann–Whitney U-test (continuous, not normally distributed variables), chi-square test or Fisher exact test (categorical variables), as appropriate.). All the p-values have been shown in the table but only those <0.05 were considered statistically significant. Title of the table has been modified and a footnote has been added to make easier the data interpretation.
The discussion is appropriate but to be certain of the right conclusions the data presentation and evaluation and especially the limits of decisions to approve or disapprove a pathological finding must be elucidated.
Following the reviewer’s suggestions, criteria for pathological findings have been elucidated.

Round 2
Reviewer 2 Report
Dear authors,
most of retinal diseases could be diagnosed in the basis of clinical examination and the existing medical documentation. I believe that stated reasons are not obstacles to the recomended procedure. But, it is good that you mentioned that in the limitations of the study.
Best wishes!
Reviewer 3 Report
Much improved, inconsistencies are removed